# Impulse Control Disorders in Parkinson’s Disease and Atypical Parkinsonian Syndromes—Is There a Difference?

**DOI:** 10.3390/brainsci14020181

**Published:** 2024-02-16

**Authors:** Mateusz Toś, Anna Grażyńska, Sofija Antoniuk, Joanna Siuda

**Affiliations:** 1Department of Neurology, Faculty of Medical Sciences in Katowice, Medical University of Silesia, 40-055 Katowice, Poland; jsiuda@sum.edu.pl; 2Department of Imaging Diagnostics and Interventional Radiology, Kornel Gibiński Independent Public Central Clinical Hospital, Medical University of Silesia, 40-055 Katowice, Poland; anna.grazynska@sum.edu.pl; 3St. Barbara Regional Specialist Hospital No. 5, 41-200 Sosnowiec, Poland; sofija.antoniuk@gmail.com

**Keywords:** impulse control disorders, Parkinson’s disease, multiple system atrophy, progressive supranuclear palsy, impulsivity, dopamine agonists, hypersexuality, compulsive buying, pathological gambling, binge eating

## Abstract

Background and Objectives: Impulse control disorders (ICDs) are characterized by potentially harmful actions resulting from disturbances in the self-control of emotions and behavior. ICDs include disorders such as gambling, hypersexuality, binge eating, and compulsive buying. ICDs are known non-motor symptoms in Parkinson’s disease (PD) and are associated primarily with the use of dopaminergic treatment (DRT) and especially dopamine agonists (DA). However, in atypical parkinsonism (APS), such as progressive supranuclear palsy (PSP) or multiple system atrophy (MSA), there are only single case reports of ICDs without attempts to determine the risk factors for their occurrence. Moreover, numerous reports in the literature indicate increased impulsivity in PSP. Our study aimed to determine the frequency of individual ICDs in APS compared to PD and identify potential factors for developing ICDs in APS. Materials and Methods: Our prospective study included 185 patients with PD and 35 with APS (27 patients with PSP and 9 with MSA) hospitalized between 2020 and 2023 at the Neurological Department of University Central Hospital in Katowice. Each patient was examined using the Questionnaire for Impulsive–Compulsive Disorders in Parkinson’s Disease (QUIP) to assess ICDs. Additionally, other scales were used to assess the advancement of the disease, the severity of depression, and cognitive impairment. Information on age, gender, age of onset, disease duration, and treatment used were collected from medical records and patient interviews. Results: ICDs were detected in 23.39% of patients with PD (including binge eating in 11.54%, compulsive buying in 10.44%, hypersexuality in 8.79%, and pathological gambling in 4.40%), in one patient with MSA (hypersexuality and pathological gambling), and in 18.52% of patients with PSP (binge eating in 3.70%, compulsive buying in 7.41%, and hypersexuality in 11.11%). We found no differences in the frequency of ICDs between individual diseases (*p* = 0.4696). We confirmed that the use of higher doses of DA and L-dopa in patients with PD, as well as a longer disease duration and the presence of motor complications, were associated with a higher incidence of ICDs. However, we did not find any treatment effect on the incidence of ICDs in APS. Conclusions: ICDs are common and occur with a similar frequency in PD and APS. Well-described risk factors for ICDs in PD, such as the use of DRT or longer disease duration, are not fully reflected in the risk factors for ICDs in APS. This applies especially to PSP, which, unlike PD and MSA, is a tauopathy in which, in addition to the use of DRT, other mechanisms related to the disease, such as disorders in neuronal loops and neurotransmitter deficits, may influence the development of ICDs. Further prospective multicenter studies recruiting larger groups of patients are needed to fully determine the risk factors and mechanisms of ICD development in APS.

## 1. Introduction

Impulse control disorders (ICDs) are a group of disorders that, according to the definition of the fifth edition of the American Psychiatric Association’s Diagnostic and Statistical Manual (DSM-5), are characterized by destructive behaviors related to impulse control, including problems with the self-control of emotions and behaviors [1]. Patients with ICDs engage in specific, repetitive activities related to the reward system, providing short-term gratification without anticipating their behavior’s personal and interpersonal consequences [2,3]. Patients do not understand or are unable to recognize their actions correctly, or being ashamed of them hides their occurrence. The repetitive, excessive, and compulsive nature of these behaviors disrupts the patient’s main areas of functioning, affecting interpersonal relationships, social and professional functioning, and everyday activities [4]. 

ICDs are an increasingly recognized non-motor psychiatric symptom in Parkinson’s disease (PD), where they include pathological gambling, compulsive shopping, hypersexuality, and binge eating. Additionally, a group of disorders that involve ICD-related behaviors (ICD-RBs) has been distinguished, such as hobbyism, excessive interest in specific activities, pathological hoarding, walkabout, and dopaminergic dysregulation syndrome (DDS) [5,6,7]. The incidence of ICDs in PD can differ vastly depending on the population and diagnostic tools used for their assessment, differing from 3.5% to 42.8% [8,9]. 

The occurrence of ICDs in PD seems to be relatively well described in the literature; however, to the best of our knowledge, in the case of atypical parkinsonian disorders (APS), there are essentially no studies on this issue. APS is a group of neurodegenerative diseases that include progressive supranuclear palsy (PSP), multiple system atrophy (MSA), corticobasal degeneration (CBD), and dementia with Lewy bodies (DLB) [10]. The incidence of APS varies depending on the disease, with a prevalence of 400 cases per 100,000 persons in the case of DLB and 1 per 100,000 in the case of CBD [11]. The aforementioned diseases were classified into the APS group due to the presence of clinical features of PD, as well as features not characteristic of classic PD, such as early-onset severe autonomic symptoms in MSA or eye movement disorders in PSP that are associated with early postural disorders and falls [12]. Accurate diagnosis of APS poses a challenge even for qualified specialists, especially in the early stages of the disease, which further complicates the assessment of individual non-motor symptoms, including ICDs [10]. Worth attention is the fact that in the case of APS, in addition to the typical symptoms for a given disease, cognitive and behavioral symptoms, e.g., depression, apathy, and anxiety, are observed from early stages [13]. The co-occurrence of increased apathy and impulsive behavior in patients with PSP, cases of patients’ predisposition to addictions in the case of MSA, as well as the occurrence of impulse control disorders, have also been demonstrated [14,15,16].

The literature contains only singular case reports of ICDs in APS; there are, however, no publications discussing this disorder in APS in a broader spectrum. Our study aims to assess the prevalence of ICDs in APS compared to PD, describe their profile, and attempt to identify their potential risk factors.

## 2. Materials and Methods

### 2.1. Subjects and Data Collection

This study covered a cohort of Polish patients with idiopathic Parkinson’s disease (PD) from the study by Toś et al. [17], extended with additional patients—a total of 185 patients with PD and 35 patients with atypical parkinsonian syndromes (27 with PSP and 9 with MSA) consecutively enrolled in the study from November 2020 to June 2023. The inclusion criteria were as follows: (a) informed and voluntary consent to participate in the study; (b) clinical diagnosis of one of the parkinsonian syndromes: PD, according to the Movement Disorder Society Clinical Diagnostic Criteria for Parkinson’s disease [18], clinically established MSA or clinically probable MSA according to the Movement Disorder Society Criteria for the Diagnosis of Multiple System Atrophy [19], and clinically probable PSP or clinically possible PSP according clinical diagnosis of progressive supranuclear palsy based on the Movement Disorder Society criteria [20]; (c) the motor and cognitive patient’s condition enabled the completion of the questionnaire. The exclusion criteria were (a) a lack of informed and voluntary consent, (b) severe dementia, and (c) diseases other than PD, MSA, or PSP. Due to the prospective nature of this study, the local ethics committee of the Medical University of Silesia approved the study (decision number PCN/0022/KB1/99/I/19/21). All of the test procedures were carried out in compliance with the ethical principles of the 1964 Helsinki Declaration and its subsequent amendments.

Patients’ demographic data (including age and gender) and medical history data were obtained from the analysis of medical records and patient interviews (including information on the time of occurrence of the first symptoms, duration of the disease, and treatment used).

### 2.2. Study Procedure and Instruments Used

The prevalence of impulse control disorders was assessed using a Polish translation of the Questionnaire for Impulsive–Compulsive Disorders in Parkinson’s Disease (QUIP) licensed by the University of Pennsylvania [21]. QUIP is a simple questionnaire completed by the patient alone or with the help of a caregiver based on the DSM-IV classification. QUIP is a commonly used and recommended questionnaire for screening for ICDs [22]. It consists of three sections containing questions (with an introductory description of the disorder) about the ICDs and the most common ICD-RBs. We adopted the cut-off points for individual ICDs in accordance with the proposal of the authors of the original validation publication as well as other authors used: (a) pathological gambling: ≥2 affirmative answers to the questions; (b) hypersexuality: ≥1; (c) compulsive buying: ≥1; and (d) binge eating: ≥2 responses [23,24].

The severity of motor symptoms was assessed by neurologists experienced in movement disorders of neurodegenerative origin with the Polish validation of MDS-Unified Parkinson’s Disease Rating Scale (MDS-UPDRS) part III for PD patients [25], the Unified Multiple System Atrophy Rating Scale (UMSARS) part II for MSA patients [26,27], and the Progressive Supranuclear Palsy Clinical Deficits Scale (PSP-CDS) for PSP patients [28]. Cognitive disorders were assessed by an appropriately trained psychologist using the mini-mental state examination (MMSE) [29] and Addenbrooke’s Cognitive Examination-III (ACE-III) [30], and depressive disorders were assessed using the Beck Depression Inventory (BDI-II) [31].

The total levodopa equivalent daily dose (LEDD), LEDD for L-dopa alone, and LEDD for dopamine agonists were counted separately to assess dopaminergic treatment’s influence and its doses on ICDs incidence [32].

### 2.3. Statistical Analysis

Statistica 13.0 software (TIBCO Software Inc., Palo Alto, CA, USA) was used for all statistical analyses. Data were reported as mean ± standard deviation (SD) for continuous data and n (%) for categorical data. The Shapiro–Wilk test was applied to assess the normality of the quantitative variables. Analysis of variance and Kruskal–Wallis tests were used to compare patients with PD, MSA, and PSP. For post hoc analysis, Scheffé’s method or Dunn’s test was used. Student’s *t*-tests, Chi-squared tests, and the Mann–Whitney U test were used to analyze the difference between patients with and without ICDs. The level of statistical significance was determined at the level *p* < 0.05.

## 3. Results

Our study group included 218 patients, 185 with PD (83.49% of all), 27 with PSP (12.39%), and 9 with MSA (4.13%). There were no differences in age and gender between the three groups. Patients with PD had an earlier age of disease onset than those with PSP (54.24 ± 10.85 vs. 64.44 ± 9.02, respectively, *p* = 0.0000), while those with MSA did not differ in age of onset from PD and PSP. PD patients had longer disease duration than those with PSP and MSA (9.59 ± 5.98 vs. 3.96 ± 1.99, *p* = 0.0000, and 9.59 ± 5.98 vs. 3.33 ± 2.12, *p* = 0.0011, respectively). Additionally, PD patients compared to PSP patients had mild or moderate dementia less frequently (8.79% vs. 18.52% and 1.10% vs. 22.22%, respectively, *p* = 0.0000). In the MSA group, no one with dementia was diagnosed.

For the treatment used, PD patients took significantly higher doses of anti-parkinsonian drugs expressed as LEDD compared to PSP patients (1026.38 ± 700.71 vs. 464.63 ± 444.41, respectively, *p* = 0.0000). Additionally, patients with PD were more likely to take DA than those with PSP (60.99% vs. 18.52%, respectively, *p* = 0.0000) and had higher LEDDs from agonists (128.78 ± 190.56 vs. 26.67 ± 73.59, *p* = 0.0003). However, there were no statistically significant differences in the frequency of use and doses of DRT between PD and MSA patients. Detailed data on the values of variables in individual diseases are presented in Table 1.

### Incidence of ICDs in Individual Diseases and Their Clinical Correlations

ICDs were present in 51 subjects (23.39% of all patients), including 45 patients with PD (24.73% of all patients with PD), 5 patients with PSP (18.52% of patients with PSP), and 1 person with MSA (11.11% of people with MSA). In PD, the most common ICDs were binge eating—in 21 patients (11.54% of all patients with PD)—followed by compulsive buying in 19 patients (10.44%), hypersexuality in 16 patients (8.79%), and pathological gambling occurred much less frequently—in 8 patients (4.40%). More than one ICD was present in 14 patients (7.69%). It was also found that hypersexuality and multiple ICDs occurred more often in men than in women (13.27% vs. 1.45%, *p* = 0.0022, and 11.50% vs. 1.45%, *p* = 0.0059, respectively). For MSA, only one ICD-positive patient met the criteria for hypersexuality and pathological gambling. For PSP, the most common ICD was hypersexuality in three patients (11.11%), followed by compulsive shopping—two patients (7.41%)—and binge eating in one patient (3.70%). Pathological gambling was not detected in any patients with PSP. However, multiple ICDs were present in one patient with PSP. There were no statistically significant differences in the frequency of ICDs between individual diseases (*p* = 0.4696) [see Figure 1].

When comparing ICD-positive and ICD-negative patients in PD, there was no significant difference in age, age of diagnosis, and gender, as well as in the severity of depression and the level of cognitive status. Patients did not differ regarding the severity of disease symptoms expressed on the H&Y scale and the MDS-UPDRS part III as well. However, we found that PD ICD-positive patients had a longer disease duration compared to PD ICD-negative patients (11.42 ± 4.72 vs. 8.98 ± 6.23, *p* = 0.0038), and were more likely to have sleep disorders (66.67% vs. 46.72%, *p* = 0.0201) and motor complications such as wearing-off phenomenon (64.44% vs. 43.07%, *p* = 0.0124) and levodopa-induced dyskinesia (64.44% vs. 40.15%; *p* = 0.0046). In terms of treatment used, as expected, PD ICD-positive patients used DA more often than ICD-negative patients (34.23% vs. 9.86%, *p* = 0.0002). ICD-positive patients used overall higher doses of anti-parkinsonian drugs expressed as LEDD (1327.53 ± 727.62 vs. 927.46 ± 665.06, *p* = 0.0018) as well as higher doses of DA (169.76 ± 115.42 vs. 115.32 ± 208.04, *p* = 0.0009) and L-dopa (986.11 ± 642.94 vs. 723.98 ± 665.06, *p* = 0.0147) [Table 2]. There was no influence of the use of other drugs or DBS on the occurrence of ICDs.

Due to there being only one MSA patient with a diagnosed ICD, this disease was excluded from the statistical comparative analysis. The patient was a 65-year-old woman who had been suffering from MSA type P for 6 years, currently treated with L-dopa monotherapy at a dose of 225 mg. The patient’s UMSARS part II score was 35; she showed signs of MCI and had mild depression.

When comparing ICD-positive and ICD-negative patients with PSP, no statistically significant differences were found in demographic data such as age, gender, duration of the disease, or age of onset. These patients did not differ in terms of the severity of the disease expressed on the PSP-CDS scale, cognitive status, severity of depression symptoms, or sleep disorders. Additionally, they did not differ statistically in terms of the treatment used and its doses [data shown in Table 3].

However, when comparing patients with PD and PSP who were ICD positive, some differences were found. PD ICD-positive patients compared to PSP ICD-positive patients had an earlier age of disease onset (53.60 ± 9.98 vs. 66.00 ± 7.25, *p* = 0.0000) and a shorter disease duration (11.42 ± 4.76 vs. 3.00 ± 1.22, *p* = 0.0000). There were no statistically significant differences in age, gender, or depression severity in the studied groups. In comparison with PSP ICD-positive patients, PD ICD-positive patients had sleep disorders more often (66.67% vs. 20%, *p* = 0.0023) and were more likely to use DA (84.44% vs. 20.00%, *p* = 0.0000) and L-dopa (93.33% vs. 80%, *p* = 0.0243). PD ICD-positive patients also used higher doses of anti-parkinsonian drugs expressed as LEDD (1327.53 ± 727.62 vs. 347 ± 287.74, *p* = 0.0000). However, PSP ICD-positive patients had more severe cognitive impairment and were more likely to suffer from moderate dementia (20% vs. 1.48%, *p* = 0.0032).

## 4. Discussion

This study aimed to assess the prevalence and severity of ICDs in different parkinsonian syndromes. We also compared the influence of several known risk factors of ICDs in those diseases. Finally, we tried to explain the origin of those symptoms in PD and atypical parkinsonism.

When comparing entire groups of patients with PD and APS without dividing them into positive and negative ICDs, we did not find any difference in the age of the subjects, which proves the high homogeneity of our study group. However, we found that patients with PD had almost three times the disease span than those with PSP and MSA, which is due to the slower progression of symptoms in PD and longer course of disease in PD than in APS [33,34,35].

Our patients with PSP had a later disease onset than those with PD, which is consistent with current epidemiological knowledge on the age of onset of these entities [36]. The more frequent and more advanced dementia in our PSP patients group compared to PD is consistent with previous research. It is related to the different pathogenesis of these diseases and the involvement of various structures within the brain [37,38]. Subjects with PD took higher doses of anti-parkinsonian drugs expressed as LEDD than those with PSP, despite the need to use higher doses of L-dopa in PSP to achieve a clinical effect, as described in previous studies [39]. This may be the result of the lower effectiveness of anti-parkinsonian drugs in PSP compared to PD and therefore the less frequent recommendation of these drugs to patients by treating physicians [40,41]. The lack of statistically significant differences in the doses of anti-parkinsonian drugs taken between PD and MSA and MSA and PSP is primarily due to the small group of patients with MSA, which limits the value of statistical calculations.

The frequency of ICD occurrence in our study among the PD patient population was 24.73%, comparable to other European populations. In a prospective ICARUS study, the Italian population exhibited an ICD frequency of 28.6% at the outset and 26.5% after two years of follow-up [8]. Similar results were found in the Spanish and French populations, with 23.48% and 25% of PD patients, respectively, displaying a spectrum of ICDs [42,43]. Moreover, our findings align with populations in Slavic countries, ethnically and culturally similar to the Polish population: in the Serbian population, the ICD frequency was 19.8% at the beginning of the study and 29.2% after a 5-year follow-up, and in the Czech population, it was 26.5% (noting that the Czech study used different diagnostic tools, namely, the South Oaks Gambling Screen and Modified Minnesota Impulse Disorders Interview, and focused exclusively on early-onset PD) [44,45,46]. While there is a considerable body of research on the occurrence of ICDs in various demographic groups of PD patients as well as distinguishing factors such as early-onset PD, very few reports address this issue in the case of APS.

Our study recognized disorders within the ICD spectrum in 18.52% of PSP patients. To date, there are no case–control studies assessing the incidence of ICDs in the population of PSP patients. Only O’Sullivan et al. and Kim et al. in their case reports described several cases of ICD spectrum disorders in the form of binge eating, compulsive shopping, hypersexuality, and excessive charity [47,48]. Moreover, few studies report on the co-occurrence of apathy and impulsiveness in PSP patients. The study by Kok et al. demonstrated that approximately 75% of PSP patients had apathy, which is consistent with apathy being the most common neuropsychiatric disorder in PSP [14]. Additionally, the researchers showed that as many as 74% of PSP patients displayed impulsive behaviors, demonstrating the highest percentage of these behaviors in the PSP patient population to date. In other studies, the prevalence of impulsiveness often ranged from 32% to 43% [49,50]. However, it should be noted that the cited authors only assessed the occurrence of impulsiveness-encompassed premature actions, without the foresight of detrimental consequences or actions made as a result of a failure to inhibit contextually inappropriate responses not coming from the ICDs themselves, and used questionnaires such as Addenbrooke’s Cognitive Examination-Revised (ACE-R), Frontal Assessment Battery (FAB), and Cambridge Behavioral Inventory-revised (CBI-R), which are not questionnaires intended for the detailed assessment of individual ICDs [51,52,53].

In the case of MSA, in our study, ICDs occurred in one out of nine patients (11.1%), and the disorders involved the simultaneous presence of pathological gambling and hypersexuality. To date, only a few case reports on the occurrence of ICDs in MSA have been published. Similarly to PSP, there are no case–control studies assessing the incidence of ICDs in MSA. In the McKeon et al. study, one case of ICDs was identified in patients with MSA, with the specific ICDs including hypersexuality and punding (watching the clock, locking and unlocking doors, and continually rearranging and lining up small objects on the desk) [54]. In Klos et al., ICDs were demonstrated in two MSA patients, both cases involving hypersexuality, which occurred after the initiation of dopamine agonist medications [55]. Cilia et al. described the case of a 65-year-old man with MSA who, after increasing his doses of L-dopa and DA, developed compulsive water drinking and a broad spectrum of other impulsive disorders such as pathological gambling and increased interest in sexual matters [15]. Tsopp et al., in their case report, described a patient who developed compulsive buying and hypersexuality during increasing the dose of piribedil and L-dopa [56]. Interestingly, in the aforementioned case reports, many patients described the occurrence of hypersexuality, which somewhat contradicts with sexual disorders often occurring in both men and women in MSA in the form of sexual frigidity or erectile dysfunction [57,58].

We found some differences in demographic data when comparing ICD-positive and ICD-negative patients in PD and PSP. First, PD ICD-positive patients had a longer disease span than ICD-negative patients. This association has been described in the literature and is probably due to the fact that patients who have been ill for longer periods of time have prolonged exposure to dopaminergic treatment and use polytherapy more often [59,60]. A similar relationship was not found when comparing ICD-positive and negative patients with PSP. This may indicate other additional mechanisms influencing the development of ICDs apart from long-term exposure to dopaminergic drugs. In our study, the only ICD-positive patient with MSA had the onset of the disease 6 years ago; in the available case reports of ICD-positive patients with MSA, the age of onset ranged from 2 to 6 years [15,55,56]. Moreover, when comparing ICD-positive patients with PD and PSP, it was found that the former has a longer disease duration. However, this is probably the result of a longer disease duration in PD patients in general.

As for other demographic factors, such as the patient’s age and the age of the first symptoms of the disease in both PD and PSP, we did not find their influence on the occurrence of ICDs. This is not reflected in most studies to date, where younger age of onset was a factor in the development of ICDs in PD [61,62]. The lack of this relationship among our subjects may result from the relatively low participation of patients with early-onset PD in our study group. As for the age of patients with PSP, in the cases described in the literature so far, the age of ICD-positive patients ranged from 66 to 83, while in our case, the average age of ICD-positive patients with PSP was 66 [47,48]. Our only patient with an ICD and MSA was 59, whereas the age of patients reported in the literature ranged from 59 to 75. Taking into account the previous literature descriptions, the relatively significant variation in patients’ age in the occurrence of ICDs in APS seems to indicate that the age of patients itself does not influence the development of ICDs. We also found no differences in gender between ICD-positive patients, both among those suffering from PD and APS. There is no clearly defined influence of gender on ICDs in the literature, as some publications indicate male gender as a risk factor for ICDs in PD, while others do not find this correlation [43,63,64]. As for patients with APS like PSP, we did not find a similar correlation as well, and in MSA, the only patient with ICD was a woman. In case reports of patients with APS, ICDs were found in both men and women. However, we found a more frequent occurrence of multiple ICDs and hypersexuality, which may result from greater sexual motivation in men and different sociocultural factors [65,66]. We found a similar but less pronounced relationship in PSP, where hypersexuality was present in three men and none of the women. Moreover, we found that PD patients with sleep disorders more often met the ICD criteria, which is consistent with previous scientific reports [67,68]. As for patients with MSA, in our single ICD-positive patient, we did not detect any sleep disorders. In PSP ICD-positive patients, we did not find a more common occurrence of sleep disorders, and there are no studies describing the impact of sleep disorders on the occurrence of ICDs in this disease. Moreover, when comparing ICD-positive patients with PD and PSP, the former suffered from sleep disorders more often. The lack of influence of sleep disorders in PSP patients on the occurrence of ICDs may be due to a different type of sleep disorders and other mechanisms of their development compared to PD [69,70]. In PSP, which is a tauopathy, the most common sleep disorders are deviated sleep architecture and insomnia, and they are more severe than those occurring in PD, which is mainly associated with more severe neurodegenerative changes in the brain stem and greater severity of orexin neurotransmission disorders than PSP [71,72,73]. In PD, however, REM sleep disorders (RBD), which are typical for synucleinopathies, are more than twice as common, which some researchers associate with a higher incidence of ICDs [74,75]. This relationship would result from the partially common pathophysiology of these two disorders with the appearance of changes in the mesocorticolimbic pathway and the limbic system [76,77,78,79].

In our study, similarly to our previous publication, we did not find any influence of depression on the occurrence of ICDs in PD, which is inconsistent with some previous publications [80]. The lack of this relationship may be partially explained by the anti-depressant effect of higher doses of DA, the use of which is a recognized risk factor for the development of ICDs. We found a similar lack of influence of depression on the occurrence of ICDs in PSP. No assessment of the severity of depression in the few published cases makes comparison impossible. Our only MSA ICD-positive patient had mild depression, while in the available case reports, the presence of depression without assessment of its severity was described only in one ICD-positive patient with MSA [55].

We also found no impact of the advancement of PD determined by H&Y and MDS-UPDRS part III on the incidence of ICDs. This is consistent with previous research results and may indicate that the motor advancement of PD itself does not influence the development of ICDs [81]. Our only MSA ICD-positive patient obtained 35 points on the UMSARS part II scale; in previous publications, this scale was used to evaluate only one MSA ICD-positive patient who received 28 points on it. We also found no impact of the PSP-CDS score on the incidence of ICDs in PSP. However, this scale focuses primarily on motor aspects of PSP, only taking into account bradyphrenia and communication disorders out of non-motor symptoms, without differentiating its causes (e.g., dysarthria, aphasia, or apraxia) [82]. However, this scale does not accurately assess other non-motor manifestations of neurodegeneration that are symptoms of PSP, such as increasing apathy or impulsivity related to the deposition of tau protein and disorders in the frontostriatal systems and front-ponto-cerebellar circuits, which, as it seems, may also contribute to the development of disorders from the ICD scope [14,83,84].

In our study, we found that PD ICD-positive patients used DA more often and at higher doses than those negative for ICDs. The relationship between the use of DA and the occurrence of ICDs is widely described in the literature [85]. It is believed that the use of dopaminergic drugs acting on D2 and D3 receptors is one of the main factors in the development of ICDs and is associated with modulating the reward network in PD patients, although the exact mechanism of this process remains unknown [86]. Some researchers suggest that taking second-generation DA may lead, especially in genetically susceptible patients, to excessive stimulation of the mesolimbic pathway, which is relatively preserved in PD, and may stimulate the development of ICDs [87,88]. This genetic susceptibility is thought to be particularly related to Ser9Gly single nucleotide polymorphism (rs6280) of the dopamine receptor D3 (DRD3) [89,90]. In terms of treatment, we also found that PD patients taking higher doses of L-dopa had a higher incidence of ICDs, consistent with previous publications [60]. Moreover, ICDs occurred most often in patients who used a combination of L-dopa and DA, which seems to confirm further its influence on the development of ICDs [91]. We found no effect of other dopaminergic drugs, such as amantadine, and COMT and MAO-B inhibitors on the development of ICDs. In previous publications, some researchers indicated a possible influence of amantadine on the development of ICDs due to its dopaminergic effects; however, other studies did not confirm this relationship, and in some cases, even found an improvement in the severity of ICDs after the use of amantadine [92,93].

Our only MSA ICD-positive patient was taking L-dopa at a low dose (LEDD 225 mg) and still developed ICD symptoms. Due to the very small group of patients with MSA, it is difficult to draw firm conclusions regarding the impact of DRT on the development of ICDs in MSA. However, based on the cases described so far, the occurrence of ICDs in MSA patients can be associated with using DRT in higher doses, especially DA. ICDs in the form of hypersexuality, binge eating, and pathological gambling developed primarily in patients using the combination of L-dopa and DA. In most of the described cases, symptoms disappeared or decreased after reducing DA doses. Based on the current research, it seems that it can be cautiously concluded that the mechanism behind the development of ICDs in MSA, which, like PD, is a synucleinopathy, may be similar to its pathomechanism and also be associated with excessive dopaminergic stimulation [94,95,96].

In our study of patients with PSP, we did not find any effect of DA and L-dopa use or their doses on the incidence of ICDs. However, only one PSP ICD-positive patient did not use any DRT, and most of them used L-dopa, which is a recognized risk factor for the development of ICDs in PD. PSP ICD-positive patients did not take anti-parkinsonian medications other than DA and L-dopa. When comparing patients that are ICD-positive in PD and PSP, the former took DA and L-dopa more often and in higher doses, but this is probably due to the fact that, in general, PD patients took higher doses of DRT than those with PSP. In the few case reports available, the authors describe the development of ICDs in PSP patients who were treated with various DAs (including pergolide, bromocriptine, rotigotine, ropinirole, and pramipexole) and L-dopa. In some of the patients described, after reducing the dosage or discontinuing DA, improvement in ICD symptoms was observed. It seems that the development of ICDs in PSP may be more complex and only partially share its pathomechanism with that present in PD. PET studies with 18 fluorodeoxyglucose showed reduced metabolism in the medial prefrontal cortex, anterior cingulate, and ventrolateral prefrontal cortex in patients with PSP [97,98]. It is believed that these brain regions may be involved in creating neural networks involved in controlling impulsive behavior, self-control, and pursuing future rewards [99,100]. Some researchers also describe the presence of structural abnormalities in the locus coeruleus, which may lead to noradrenergic deficits and thus disorders of both motor and cognitive control, including impulsivity with a different pattern than in PD [101]. Therefore, it seems that the increased impulsivity described by many researchers in patients with PSP resulting from disorders within neuronal loops and neurotransmitter deficits combined with excessive dopaminergic stimulation may lead to an increased risk of developing ICDs [14].

Our study, to the best of our knowledge, is the first work that undertook a prospective assessment of the occurrence of ICDs in the population of patients with APS and aimed to demonstrate similarities and differences in terms of risk factors and potential pathomechanisms compared to the group of PD patients. We managed to establish that ICDs occur not only in PD patients but can also be quite frequently present in APS patients. This is particularly important due to the potentially very harmful impact of ICDs on family and social lives. There have been reports of dangerous sexual behaviors directed toward close individuals, which could have tragic outcomes [102,103]. Therefore, paying attention to ICDs in both PD and APS patients is crucial for appropriate treatment and maintaining good relationships between patients, their families, and friends. We are aware that our study had certain limitations. Firstly, it was a single-center study, resulting in a relatively small number of patients eligible for the study, especially in the case of the group of patients with atypical parkinsonism. Atypical parkinsonian syndromes are rare, occurring much less frequently than Parkinson’s disease. Therefore, comparing these two groups poses a challenge, affecting the statistics and results. Due to the rarity of atypical parkinsonism, there are no research articles on the occurrence of ICDs in patients with atypical parkinsonism, and only individual case studies are available. The limited representation of studies on ICDs in APS may also stem from the fact that ICDs are a group of disorders that clinicians may ignore amidst the primary symptoms of these medical conditions. Consequently, these symptoms may go unnoticed by the patients, significantly impacting their quality of life as well as their caregivers and families. Another limitation of our study is the need to validate the QUIP scale for patients with atypical parkinsonism. This is due to the diversity within the spectrum of diseases in the atypical parkinsonism group and the frequency of their occurrence. Our work will contribute to highlighting this issue and help create validation for one of the scales assessing ICDs for the group of patients with atypical parkinsonism. However, this requires multicenter studies.

## 5. Conclusions

ICDs are well known non-motor disorders in PD. However, their presence can often be omitted in routine clinical visits, which is particularly important due to their potentially harmful impact on patients’ lives. The risk factors for ICDs in PD are becoming better known and confirmed in numerous studies. This allows for an increase in awareness among doctors and patients about the possibility of developing ICD symptoms. However, apart from single case reports, there are no broader studies on the occurrence and risk factors of ICDs in APS. Such studies are difficult due to the much less common occurrence of APS compared to PD. Our study suggests that ICDs may occur with comparable frequency in both PD and APS, affecting a substantial portion of patients afflicted with these conditions. Consistent with findings in other European cohorts, the most prevalent ICDs in PD were identified as binge eating, compulsive buying, and hypersexuality. In MSA, the sole patient with ICDs manifested symptoms of hypersexuality and pathological gambling, whereas in PSP, the predominant ICDs observed were hypersexuality, compulsive shopping, and lastly, binge eating. We confirmed widely reported risk factors for ICDs in PD, such as longer disease duration, sleep disorders, and dopaminergic treatment, while simultaneously not observing these risk factors in APS. Based on our study findings and previous case reports, it appears that the risk of developing ICDs in APS may involve additional factors that require further identification in addition to classical risk factors such as dopaminergic treatment. Some of these factors may be directly related to the pathomechanisms of APS themselves. This especially applies to PSP, a tauopathy, where an increased level of impulsiveness is observed. Further multicenter and multicountry studies are recommended to enable the recruitment of sufficiently large study groups, which will allow for the determination of certain risk factors for ICDs in APS and potentially provide better insight into the mechanisms of their formation.

## Figures and Tables

**Figure 1 brainsci-14-00181-f001:**
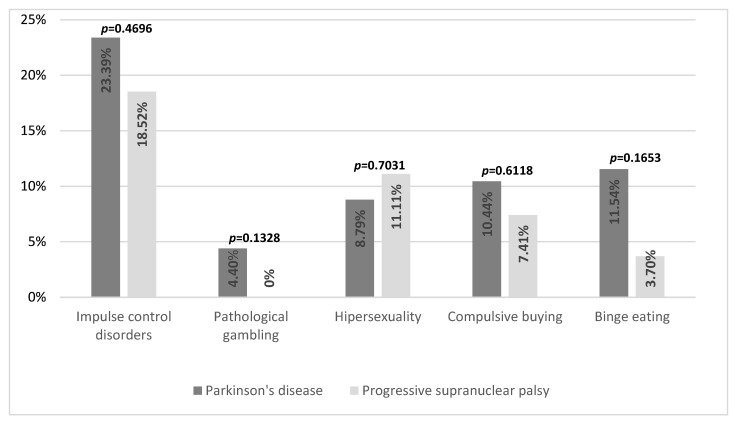
Frequency of impulse control disorders in Parkinson’s disease and progressive supranuclear palsy.

**Table 1 brainsci-14-00181-t001:** Clinical features of the study population.

	Parkinson’s Disease	Progressive Supranuclear Palsy	Multiple System Atrophy	*p*	Post Hoc Comparisons
N:	182 (83.49%)	27 (12.39%)	9 (4.13%)		
Gender:	M 113 (62.09%)	M 15 (55.56%)	M 2 (22.22%)	0.0530	
	F 69 (31.91%)	F 12 (44.44%)	F 7 (77.78%)		
Age at examination (years)	63.88 ± 9.47	68.44 ± 8.19	64.00 ± 8.72	0.0793	
Age at onset (years)	54.24 ± 10.85	64.44 ± 9.02	60.67 ± 7.63	0.0000 *	PSP > PD
Disease duration (years)	9.59 ± 5.98	3.96 ± 1.99	3.33 ± 2.12	0.0000 *	PD > PSP, PD > MSA
Depressive disorders:				0.0707	
Minimal depression	152 (83.51%)	22 (81.48%)	5 (55.56%)		
Mild depression	23 (12.64%)	2 (7.41%)	4 (44.44%)		
Moderate depression	7 (3.85%)	2 (7.41%)	0 (0.00%)		
Severe depression	0 (0.00%)	1 (3.70%)	0 (0.00%)		
Dementia:				0.0000 *	PSP > PD
No dementia	102 (56.04%)	6 (22.22%)	4 (44.44%)		
MCI	62 (34.07%)	10 (37.04%)	5 (55.56%)		
Mild dementia	16 (8.79%)	5 (18.52%)	0.0000		
Moderate dementia	2 (1.10%)	6 (22.22%)	0.0000		
Sleep disorders	94 (51.65%)	8 (29.63%)	5 (55.56%)	0.0945	
Dopamine agonist use	111 (60.99%)	5 (18.52%)	3 (33.33%)	0.0000 *	PD > PDP
DA-LEDD (mg):	128.78 ± 190.56	26.67 ± 73.59	52.22 ± 78.39	0.0003 *	PD > PSP
Levodopa:	158 (86.81%)	20 (74.07%)	8 (88.89%)	0.1799	
LD-LEDD (mg):	789.51 ± 580.06	397.22 ± 414.95	591.67 ± 391.91	0.0013 *	PD > PSP
Total LEDD (mg):	1026.38 ± 700.71	464.63 ± 444.41	709.44 ± 471.92	0.0000 *	PD > PSP

Data are shown as numbers and percentages for qualitative variables and mean ± SD for quantitative variables. *—statistically significant differences; PSP—progressive supranuclear palsy; PD—Parkinson’s disease; MSA—multiple system atrophy; M—male; F—female; MCI—mild cognitive impairment; DA—dopamine agonist; LEDD—levodopa equivalent daily dose; LD—levodopa.

**Table 2 brainsci-14-00181-t002:** Association of clinical features and ICDs in patients with Parkinson’s disease.

	ICD Patients	Non-ICD Patients	*p*
Male	71.11%	59.12%	0.1504
Age at examination (years)	65.00 ± 9.15	63.51 ± 9.57	0.3671
Age at onset (years)	53.60 ± 9.98	54.45 ± 11.15	0.6488
Disease duration (years)	11.42 ± 4.76	8.98 ± 6.23	0.0038 *
MDS-UPDRS part III “OFF state”	39.75 ± 16.41	42.44 ± 19.59	0.6548
MDS-UPDRS part III “ON state”	19.45 ±12.40	20.91 ± 13.06	0.6886
Motor fluctuations	82.22%	50.37%	0.0002 *
Depressive disorders:			0.2238
Minimal depression	86.67%	82.48%	
Mild depression	6.67%	17.52%	
Moderate depression	6.67%	2.92%	
Severe depression	0.00%	0.00%	
Dementia:			0.7234
No dementia	60.00%	54.74%	
MCI	28.89%	35.77%	
Mild dementia	8.89%	8.76%	
Moderate dementia	0.00%	0.73%	
Sleep disorders	66.67%	46.72%	0.0201 *
Dopamine agonist use	84.44%	53.52%	0.0002 *
DA-LEDD	169.76 ± 115.42	115.32 ± 208.04	0.0009 *
MAO-B inhibitors	24.44%	16.79%	0.2529
Amantadine	33.33%	26.28%	0.3605
Levodopa	93.33%	84.67%	0.1604
LD-LEDD	986.11 ± 642.94	723.98 ± 665.06	0.0147 *
Total LEDD	1327.53 ± 727.62	927.46 ± 665.06	0.0018 *

Data are shown as numbers and percentages for qualitative variables and mean ± SD for quantitative variables. *—statistically significant differences; DA—dopamine agonist; LEDD—levodopa equivalent daily dose; LD—levodopa; MDS-UPDRS—MDS Unified Parkinson’s Disease Rating Scale.

**Table 3 brainsci-14-00181-t003:** Association of clinical features and ICDs in patients with progressive supranuclear palsy.

	ICDs Patients	Non-ICDs Patients	*p*
Male	80.00%	50.00%	0.2068
Age at examination (years)	69.20 ± 6.42	68.27 ± 8.66	0.8243
Age at onset (years)	66.00 ± 7.25	64.09 ± 9.49	0.6678
Disease duration (years)	3.00 ± 1.22	4.18 ± 2.08	0.2612
PSP-CDS	8.40 ± 1.95	8.68 ± 2.63	0.8998
Sleep disorders	20.00%	31.82%	0.5904
Depressive disorders:			0.5241
Minimal depression	80.00%	81.82%	
Mild depression	20.00%	9.09%	
Moderate depression	0.00%	4.55%	
Severe depression	0.00%	4.55%	
Dementia:			0.0673
Lack of dementia	60.00%	16.64%	
MCI	0.00%	45.54%	
Mild dementia	20.00%	18.18%	
Moderate dementia	20.00%	22.73%	
Levodopa	80.00%	72.73%	0.7321
LD-LEDD	315.00 ± 234.25	415.91 ± 448.05	0.8271

Data are shown as numbers and percentages for qualitative variables and mean ± SD for quantitative variables. PSP-CDS—Progressive Supranuclear Palsy Clinical Deficits Scale; LEDD—levodopa equivalent daily dose; LD—levodopa.

## Data Availability

The datasets generated and analyzed during the current study are available on request from the corresponding author. The data are not publicly available due to privacy reasons.

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
