# Peer review of "Impulse Control Disorders in Parkinson’s Disease and Atypical Parkinsonian Syndromes—Is There a Difference?"

_brainsci, 2024, doi:10.3390/brainsci14020181_

Round 1
Reviewer 1 Report
Comments and Suggestions for Authors
The paper is devoted the actual problem which remainded in the dark for a long time despite the fact the doctors often dealt with it, not only in Parkinson disease, but in atypical parkinsonism too. This is one of the few Studies that systematically investigated CD in various form of Parkinsonism.
Comments on the Quality of English Language
Language is good enough
Author Response
Dear Reviewer,
We greatly appreciate your recognition of the importance of the topic we have addressed and your kind and positive evaluation of our article.
Reviewer 2 Report
Comments and Suggestions for Authors
The main question under consideration is the difference between impulse control disorders in Parkinson’s disease and atypical parkinsonian syndromes.
The topic is relevant and addresses an important issue in Parkinson's disease research - impulse control disorders. The aim of the study under review is to assess the prevalence of impulse control disorders in atypical parkinsonian syndromes compared to Parkinson’s disease, describe their profile, and attempt to identify their potential risk factors.
This research provides valuable insight into the differences and risk factors for people with Parkinson’s disease and atypical parkinsonian syndromes.
The methodological part of this paper is well structured and clear. No issues were found in terms of methodology improvement.
The conclusions are consistent with the data presented but they are more about describing the current scientific area and future studies. I’d advice the authors to include some sentences about the results of their own study.
The references seem to be appropriate.
In summary, the study is well conducted and provides insightful information. Improvements in the Conclusions section could have improved the presentation of the study results.
Author Response
Dear Reviewer,
thank you for your positive assessment of our article and for your valuable and constructive comments and suggestions that will help us make it better. Below are point-by-point answers to your evaluation.
“The conclusions are consistent with the data presented but they are more about describing the current scientific area and future studies. I’d advice the authors to include some sentences about the results of their own study.”
According to the suggestion, we supplemented the conclusion section with information from our results.
Reviewer 3 Report
Comments and Suggestions for Authors
This seems to be a very well executed and presented study on ICDs in atypical Parkinsonism. It is a novel study in that no-one appears to have reported rates of occurrence of ICDs in this group before. As this seems to be important data that should be available in the public domain as soon as possible I do not have any recommendations for revision.
Comments on the Quality of English Language
Could do with a check through for typos (e.g. “hipersexuality”) and improvements to clarity of writing in parts.
Author Response
Dear Reviewer,
thank you for appreciating our article and the importance of the topic we addressed, as well as for the valuable suggestions that will enhance the quality of our article. We have included a response to your evaluation below.
“Comments on the Quality of English Language
Could do with a check through for typos (e.g. “hipersexuality”) and improvements to clarity of writing in parts.”
As suggested, we removed typos and improved the overall clarity of the text.